# Research on Factors Affecting Solvers' Participation Time in Online Crowdsourcing Contests

**Keng Yang**

School of Information, Central University of Finance and Economics, 39 South College Road, Haidian District, Beijing 100081, China; 2016110134@email.cufe.edu.cn

**Abstract:** A crowdsourcing contest is one of the most popular modes of crowdsourcing and is also an important tool for an enterprise to implement open innovation. The solvers' active participation is one of the major reasons for the success of crowdsourcing contests. Research on solvers' participation behavior is helpful in understanding the sustainability and incentives of solvers' participation in the online crowdsourcing platform. So, how to attract more solvers to participate and put in more effort is the focus of researchers. In this regard, previous studies mainly used the submission quantity to measure solvers' participation behavior and lacked an effective measure on the degree of participation effort expended by a solver. For the first time, we use solvers' participation time as a dependent variable to measure their effort in a crowdsourcing contest. Thus, we incorporate participation time into the solver's participation research. With the data from Taskcn.com, we analyze how participation time is affected four key factors including task design, task description, task process, and environment, respectively. We found that, first, for task design, higher task rewards will attract solvers to invest more time in the participation process and the relationship between participation time and task duration is inverted U-shaped. Second, for task description, the length of the task description has a negative impact on participation time and the task description attachment will positively influence the participation time. Third, for the task process, communication and supplementary explanations in a crowdsourcing process positively affect participation time. Fourth, for environmental factors, the task density of the crowdsourcing platform and the market price of all crowdsourcing contests have respectively negative and positive effects on participation time.

**Keywords:** crowdsourcing contests; open innovation; participation time; task characteristics; participation behavior

---

## 1. Introduction

Crowdsourcing is the act of an enterprise that outsources the work traditionally performed by internal employees through an online public call to an uncertain group of people without clear boundaries [1]. Crowdsourcing contests are one of the most popular and effective crowdsourcing models [2]. A crowdsourcing contest mainly consists of three elements, as follows: Seekers, solvers, and the intermediary platform. The typical process is that a seeker sets the task requirements through the crowdsourcing platform and the solvers choose to participate and submit the solutions according to their own skills and knowledge. Then, the solver who provides the best solution will win the rewards [3]. There are some famous crowdsourcing platforms such as InnoCentive.com and Kaggle.com from the U.S., as well as Taskcn.com and ZBJ.com from China, etc. In a crowdsourcing contest, any participant can submit a task solution, but usually only the participant who provides the highest quality solution will win (i.e., winner take all) [4].

The performance of a crowdsourcing contest mainly depends on the solvers' active participation and effort. The solvers are completely voluntarily involved in the crowdsourcing contest and

whether the solvers actively participate in the contest is crucial to the survival of the crowdsourcing platform [5]. In the extant research, the main indicators to measure the performance of a crowdsourcing contest are the following: The number of submissions, the average quality of submissions, and the quality of the winning submission [6]. Yang et al. [7] used the number of submissions to measure participants' behavior and concluded that a crowdsourcing contest that has higher rewards, a longer duration, a shorter description, a lower time cost, and a higher popularity will attract more solvers. Leimeister et al. [8] believe that the higher the winning bidder's rating, the more likely the bidder is to propose an optimal solution. Girotra et al. [9] argued that the quality of the best creative ideas is determined by four factors including the average quality of ideas generated, the number of ideas generated, the variance in the quality of ideas generated, and the ability of the group to discern the quality of the ideas. Walter and Back [10] used the average quality of solutions and the number of solutions to measure the performance of a crowdsourcing contest and they examined the impact of rewards, duration, and the length of task description on the performance of the crowdsourcing contest. They found that the quantity and quality of solutions were affected by different types of external factors. So, if seekers try to maximize the quantity and quality of solutions by increasing task rewards, some conflicts may arise and the quantity and quality goals cannot be achieved at the same time. The seekers' purpose in online crowdsourcing contests is to attract high-quality solutions and to obtain good and diverse solutions, which not only depends on task design but also on the seekers' understanding of the motivation and behavior of the solvers.

The effort invested by a solver determines the quality of his/her solution and, then, the quality of his/her solution will have an impact on the probability of his/her winning. So, the participation effort is significantly related to the probability of winning. Usually, the level of participation effort can be measured objectively by participation time. Some researchers regard participation time as an independent variable in order to examine its impact on the probability of winning of a solver. Yang et al. [11] found that the longer the time interval between registration and submission, the higher the chance of winning. Mahr et al. [12] believe that winning a solution depends on the solver's time investment and familiarity with the task. Bockstedt et al. [13] measured the solvers' participation behavior in three dimensions of solvers, the location where the solver first submitted, number of solvers, and solvers' participation time. The results show that increasing the participation time of a solver has a positive impact on his/her winning. Participation time directly reflects the time cost of the solver's input in the task process, and also reflects the level of participation effort of solvers. Solvers with longer participation time will spend more time on designing projects and will have the opportunity to get more information about the task and other solves, which will make the solution submitted by solvers better.

According to the existing research, the number of submissions is used to measure solvers' participation behavior in most research and the research on the participation time is relatively limited. If any, they only do the research from the perspective of solvers and study how participation time affects the probability of winning. The number of submissions reflects the number of solvers who decide to participate in a crowdsourcing contest and the participation time reflects the degree of the solvers' participation effort. For seekers, they hope for more solvers and various solutions from which to choose. In addition, they want to motivate solvers to invest more effort into providing solutions. However, the measurement is used to be very subjective. Therefore, for the first time, we used participation time as the dependent variable to measure the level of participation effort of the solvers and to analyze the factors that affect solvers' participation time.

## 2. Literature Review

From the existing research, there are many factors that influence the solvers' behavior in crowdsourcing contests. We divide these influencing factors into four categories, as follows: Task design, task description, task process, and environmental factors. The task rewards and task duration in the process of task design are important factors for many scholars. Task rewards are the motivation

for solvers to gain economic value and compensate for their costs and energy. Rich et al. [14] found that the increase in task rewards can stimulate solvers' motivations and improve their participation effort. Liu et al. [15] divided the participation process in crowdsourcing contest into two phases, the registration phase and the solution submission phase. They found that the higher the task rewards, the higher the participation rate in the two phases. Based on the theory of expectation to model factors that influence the solvers' participation effort, Sun et al. [16] found that task rewards and trust have a positive impact on participation efforts. Li and Hu [17] have empirically found that task rewards are positively correlated with the number of registrations and submissions. Ye and Kankanhalli [18] found that monetary rewards, skills upgrading, job autonomy, enjoyment, and trust will positively affect solvers' participation in crowdsourcing contest. Dipalantino and Vojnovic [4] analyzed the relationship between participation and incentives, pointing out that task rewards are not the only factor affecting participation, and that the impact of the task duration should also be considered. Yang et al. [7] found that higher task rewards, longer task durations, and shorter task descriptions would attract more solvers.

Task descriptions are the specific requirements of seekers for solvers' solutions. The quality of the task description directly affects the certainty of a task to be solved, thus affecting the solvers' participation behavior. Yang et al. [7] studied online crowdsourcing contests through large-scale data and found that shorter task descriptions would attract more solutions, especially for tasks based on simple ideas. Finnerty et al. [19] implemented two sets of experimental studies to conclude that task rewards are the factors that motivate solvers to produce higher quality solutions and that clear and simple task descriptions can help improve crowdsourcing performance. By studying the impact of task description length and task duration on the success of the project in the software crowdsourcing contest, Gefen et al. [20] found that the longer the task description and the longer the task duration of the project, the more likely it is to succeed. In addition, Zhou et al. [21] studied the impact of project descriptions of crowdfunding in a crowdfunding platform, which is similar to crowdsourcing. They found that the length and the readability of the description significantly affected the probability of crowdfunding success. Jiang et al. [22] studied the impact of problem specification of task description on the outcome of the competition and found that if more conceptual objectives are disclosed in the problem specification, the number of solvers in the competition decreases, but the workload of each solver does not change. If more implementation guidelines disclosed in the problem specification, each solver's workload increases, but the number of solvers in the game does not change. According to the uncertainty reduction theory, Pollok et al. [23] believe that a clear and concise problem description can increase the confidence of potential solvers and thus increase their willingness to participate.

Task communication during the task process will also affect the solvers' participation. Huang et al. [24] found that a larger number of feedbacks and faster feedback speeds from enterprises in the crowdsourcing process can significantly increase the number of creative submissions. Wooten and Ulrich [25] compared the impact of no feedback, random feedback, and directional feedback on the participation of solvers through a field experiment on the crowdsourcing contest platform and found that the directional feedback helps to improve the average quality of the solutions. Jian et al. [26] found that the amount of feedback in the crowdsourcing process had a positive impact on the number of solutions submitted. Jiang et al. [27] empirically tested the impact of feedback on the results of the crowdsourcing contest and found that feedback positively affects solvers' participation. Mihm and Schlapp [28] discussed how organizers of a contest can use the feedback policy to design the best information structure to improve the contest performance. It is believed that the lack of feedback on the task will cause some specific information to be ignored by the solvers and, thus, will fail to achieve the expected performance.

The competitive environment factors are the external factors and objective factors that affect the solvers' participation. Generally, the competitive environment factor is measured by task density and market price (task density is also called competition intensity in some literature, which is collectively referred to as task density in this paper). Shao et al. [29] used the number of solvers and the level

of solvers who win the bid to reflect solvers' participation behavior and analyzed the effects of task attributes (including task rewards, task duration, and task difficulty) and environment factors (including task density and market price) on participation behavior. They concluded that higher rewards, easier tasks, and longer durations will lead more solvers to participate. Then, they used task density to indicate the number of other tasks in the same category within a certain number of days and found that the task with a higher task density will attract fewer solvers. Additionally, they used the average price of other competing tasks during the same period to represent the market price, which was found to have no significant effect on the number of solvers. Dissanayake et al. [30] studied the impact of the social and intellectual capital allocation of members of virtual teams on team performance in online crowdsourcing contests and found that, when the competition is fierce, social and intellectual capital adjustments have a negative impact on team performance. When there is less competition, this adjustment has a positive impact on team performance.

In addition, some scholars have done some influence mechanism research from the perspective of participation motivation and found that extrinsic motivation (such as adding monetary rewards) usually strengthens the willingness of solvers' to accept a task and increases the speed at which tasks are completed, while intrinsic motivations can improve output performance [31]. Some studies focus on the solvers' level of participation effort. Zhao and Zhu [32] analyzed the effects of various motivations on participation behavior and found that external motivation in the crowdsourcing contest is positively related to the level of participation effort. Based on the theory of expected value, Sun et al. [16] conducted a nonlinear analysis of the relationship between incentives and efforts and found that task rewards had a positive impact on efforts. Liang et al. [33] studied how intrinsic and extrinsic motivation affects solvers' participation effort in crowdsourcing contests and found that there is a negative adjustment between intrinsic motivation and extrinsic motivation, indicating that when the level of extrinsic motivation is higher, the effect of intrinsic motivation on the task will be diminished and extrinsic incentives actively adjust the relationship between participation motivation and effort. Lorenzo-Romero and Constantinides [34] analyzed the motives of customers to participate in online collaborative innovation processes and they found that financial motives are not the main reason for co-creation; highly motivated customers are motivated by product-related benefits, while hedonic benefits are the most important triggers for less motivated co-creators. However, the research on the participation motivation and effort of solvers is very subjective and lacks effective and objective measurement.

Through the above literature review, we identified that many factors will affect solvers' participation behavior. Nonetheless, in past research, solvers' participation behavior was mainly measured by the number of solution submissions and the research on the solvers' participation time is insufficient. In this paper, we empirically examine what factors affect solvers' participation time in crowdsourcing contests.

## 3. Development of Hypotheses

According to the literature review above, it is believed that the factors affecting solvers' participation time include task design, task description, task process, and environmental factors. We constructed an affecting model of participation time and the affecting mechanism is in Figure 1.

Participation time indicates the time invested by a solver in the completion of the task, which reflects the effort of solvers who chose to participate in the task. The task rewards not only affect the number of submissions, but also affect the level of effort expended by the solvers after choosing to participate. The expected value theory is mainly used in research to explain the effects of task rewards on solvers' participation. Shepperd [35] summarizes expected value as a function of expectations, media, and value. He thought that a higher probability of winning contributes to a higher expected value, which will increase the participation motivation of solvers and attract them to expend more participation effort. So, in order to improve their expected value, the solver who participates in a crowdsourcing contest will invest more time and energy to improve the quality of solution and increase

the attention to the task, so as to increase the probability of receiving monetary rewards. In another way, increasing task rewards will further incentivize solvers' motivation and strengthen the participation effort of solvers [14]. Solvers have an extrinsic motivation to pursue higher economic returns, which drives the solvers to continue to focus on the task and invest more time and energy [16].

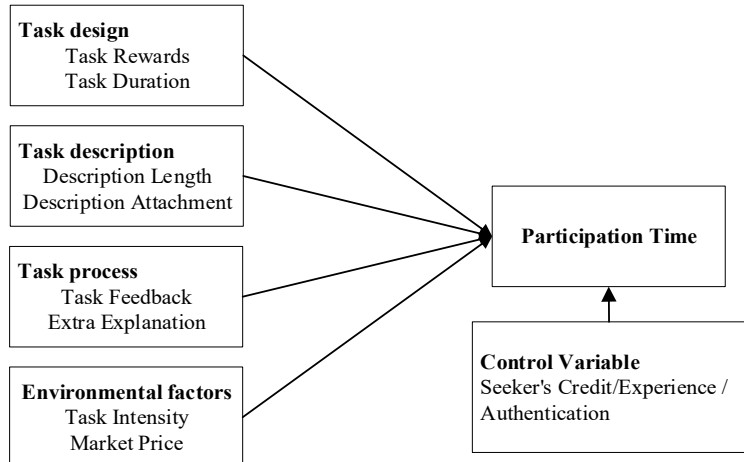

**Figure 1.** Participation Time Research Model.

Thus, we use participation time to indicate the degree of participation effort and believe that the solvers in the participation state will invest more time and cost to obtain higher expected returns by increasing the probability of successful biding. The participation time is the working time that the solver choses to participate in the task and expend his/her effort. The more the investment time, the greater the effort and the expected returns. So, we believe there is a positive linear relationship between participation time and task rewards and proposed the first Hypothesis 1a.

**Hypothesis 1a.** *Task rewards have a positive impact on the solvers' participation time.*

Task duration is also an important factor that researchers focus on. According to the expected value theory, for solvers, the longer the task duration, the more time the solvers get to complete the task. Thereby, they can improve the quality of a solution to gain the expected value. Some studies have shown that longer task durations will result in better performance of crowdsourcing contests [10,20]. The solver's participation in the crowdsourcing contest is a kind of trading behavior and his/her ultimate goal is to obtain the highest expected value at the lowest cost. Hence, to recover the input cost in the short term is one of the important factors for a rational solver to consider. For a task with a short duration, solvers can obtain the bidding result as soon as possible, can achieve the expected value with a certain probability, and this will contribute to cover solvers' cost in a short time. If the duration is too long, the participation cost will increase, so that the time for the successful bidder to obtain the task rewards becomes long and it will require solvers to expend more time and pay more attention. Korpeoglu et al. [36] believe that a longer task duration will increase the workload of each solver, improve the quality of task solutions, and increase the return of the seeker. However, with the task duration increasing, solvers' participation will be decreased and in turn, the revenue of seekers will be discounted. Since, for solvers, a longer task duration means a longer cost-recovery period and rising opportunity cost, then the solver's patience will be eliminated which makes it unnecessary for him/her to invest too much time to complete a task with a lower expected value, his/her participation time is reduced. Therefore, when the task duration exceeds a certain threshold, solvers will not invest too much time to complete the task. We believe that the task duration has an inverted U -shaped relationship with participation time, and Hypothesis 1b is proposed as follows:

**Hypothesis 1b.** *Task duration has an inverted U-shaped effect on the solvers' participation time.*

Task description is the expression of the specific requirements of seekers for task solutions. Task descriptions help the solvers understand the task, analyze the task, and decide whether to participate in the task. Yang et al. [7] found that a shorter task description will attract more solutions, especially for projects based on simple ideas that are more effective in obtaining solutions. The clearer the information conveyed by the task description, the higher the quality of the information, the more specific the requirements, and the more objective it is for the solver to assess whether he is competent for the task increases the willingness to participate. However, if the length of the description is too long, it may have the opposite effect, which makes solvers spend more energy and time cost to analyze the task and make judgments, which dampens their participation enthusiasm and is not conducive to the improvement of crowdsourcing contest performance [37].

A task attachment is a description provided by a seeker as an attachment when the task is described, usually a picture or a document. Task attachments help improve the readability of task descriptions and give solvers more specific task information. Therefore, solvers will be willing to invest more time and effort to complete the task. Based on this, we propose Hypothesis 2a and 2b, as follows:

**Hypothesis 2a.** *The length of the task description has a negative impact on the solvers' participation time.*

**Hypothesis 2b.** *The attachment of the task description has a positive impact on the solvers' participation time.*

Numerous studies have shown that the more communication there is between seekers and solvers on the task, the more accurate the solvers' understanding of the task and the more motivated the solver is to participate in the task. Huang et al. [24] studied the impact of corporate feedback behavior on crowdsourcing performance and found that increasing the number of communications and the communication speeds significantly increased the number of creative submissions. Communication and supplemental instructions during the task process establish communication channels between seekers and solvers so that solvers can receive intermediate comments from seekers and modify their solutions accordingly. Jiang et al. [27] found that the feedback of seekers in the course of the task process positively affected solvers' participation. Task communication can reduce the information asymmetry between seekers and solvers, thereby reducing the risk of solvers' effort in the wrong direction and reducing the cost of submissions expected by solvers. During the task process, seekers can supplement the released tasks and the supplementary explanation will enhance solvers' further understanding of the task, which is helpful for solvers to work in the right direction. Accordingly, we propose Hypothesis 3a and 3b, as follows:

**Hypothesis 3a.** *Task communication in the task process has a positive impact on the solvers' participation time.*

**Hypothesis 3b.** *The supplementary notes in the task process have a positive impact on the solvers' participation time.*

Task density reflects the competition intensity of different tasks over a period of time. The more similar tasks, the fewer solvers will participate in the same task [29]. The greater the task density, the more tasks the solvers can choose to participate in, and the time spent on each task will be reduced.

The market price reflects the average price of similar tasks on the platform during the same period. When the market price is high, this means that other competitive projects set higher rewards. When a solver chooses to participate in a project, the solver faces a higher opportunity cost for the loss of other higher rewards [7]. Expected value theory holds that a person's behavior is related to the expected value of the behavioral payoff, the probability of winning, and achievement [38]. Shah and Higgins [39] argue that incentives such as task rewards to motivate participants' behavior usually depend on the participant's expected value of getting rewards. When the expected value is high, even if the task rewards themselves are small, the incentive effect of task rewards on solvers will increase, that is, the enthusiasm and participation effort of solvers will increase. Wigfield et al. [40] found that

what really works is the expected value, not the actual task rewards. If the expected value is low, even if the task rewards are high, the enthusiasm of the solvers to participate in the task will decline. When the market price is high, in order to make up for the higher opportunity cost, solvers will put in more effort to increase the probability of winning, that is, they will increase the participation time to increase the expected value. Accordingly, Hypothesis 4a and 4b are proposed as follows:

**Hypothesis 4a.** *The task density of environmental factors has a negative impact on the solvers' participation time.*

**Hypothesis 4b.** *The market price of environmental factors has a positive impact on the solvers' participation time.*

## 4. The Data and Measurement

### 4.1. Sample Selection

The data in this paper is from Taskcn.com, which is one of the biggest online crowdsourcing platforms in China and was founded in 2006. The involved competition tasks included design, writing, and programming. As of January 2019, its registered users exceeded 3.70 million, tasks exceeded 65,000, the total amount of task rewards amounted to 40 million yuan (RMB), and the number solvers who get rewards by winning has exceeded 274,897.

We used a web crawler tool written by the Python program to capture all the tasks that were released under the reward module and a total of 4011 tasks, completed between 1 February 2015 and 1 January 2018, were obtained. These tasks were released by 2881 seekers and 25,951 solvers participated in these crowdsourcing contests. We chose the single-person winning mode as the research sample and dropped some abnormal samples. Finally, we got 2500 pieces of valid data.

### 4.2. Variable Definition

Dependent variable: The dependent variable was participation time. Participation time indicates the average time lapse between entering time and submission time. The stages of the crowdsourcing contest process are shown as follows in Figure 2.

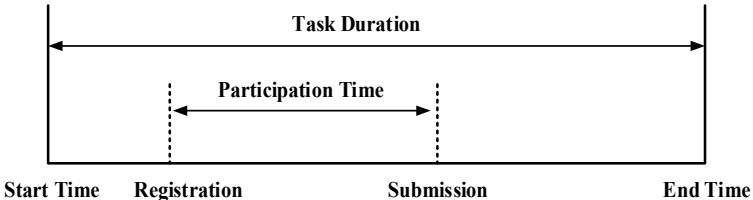

**Figure 2.** The stages of the crowdsourcing contest process.

Independent variable: Task rewards are the rewards set by the task seeker for the bid winner. Task duration indicates the duration of the task opening, set by seekers, from the start of the task to the end of the task. Task description length is the number of characters used in the task description. Description attachment is the picture or document attachment provided by seekers when describing the task. Task communication uses the number of communications between seekers and solvers during the task process. A supplementary explanation is the further description of the task. Task density is the number of similar tasks in a given period of time in the same crowdsourcing platform and market prices are the average price of similar tasks over a period of time.

Control variables: The characteristics of seekers themselves will also affect the participation behavior of solvers, including seekers' credit, experience, favorable rate, and real-name authentication. In addition, in order to prevent the influence of sample data fluctuations with time on the research results, we add the monthly dummy variable as part of the control variables in the regression model to control the uncertainty of the sample data over time. The task type and the winning mode were used

in the data filtering process and are no longer present as control variables. The variables used in this study are clearly defined in Table 1.

For the convenience of regression, we refer to the treatment way of variables used by Shao et al. [29] and Li and Hu [17]. In this study, the variables of continuous non-normal distribution are logarithmically processed and logarithmic processing does not affect the relationship between variables.

**Table 1.** Description of the definition of the main variables of the model.

| Variable Types | Variable Name | Definition |
|---|---|---|
| Dependent Variables | Participation time | The simple arithmetic means of the time spent by all solvers on solving the task, that is, the average participation time, referred to as participation time, in minutes. |
| Independent Variables | Task rewards | Seeker sets the monetary reward for the winner of the task, is measured in RMB. |
| | Task duration | The period between start time and end time of task set by seekers is used as a measurement for project duration, is measured in days. |
| | Description length | The number of characters used in the task description. |
| | Description Attachment | The picture or files provided by seekers for task description. The value 1 indicates the task including attachment, otherwise the value is 0. |
| | Task Feedback | The number of communications between seekers and solvers in the task process. |
| | Extra Explanation | The task supplementary explanation added by seekers in the task process. In Taskcn.com, the supplementary explanation is independent of the task description and is displayed in an additional position below the task description. |
| | Task Density | The task density is the number of similar tasks for 7 days before and after the start time for a task. |
| | Market Price | The market price is the average price of other similar tasks for 7 days before and after the start time for a task. |
| Control Variables | Seeker's Credit | The seeker's credit in Taskcn.com. |
| | Seeker's Experience | The number of tasks that seeker released in Taskcn.com. |
| | Seeker's Authentication | If the seeker has real-name authentication, which is 1, otherwise 0. |
| | Time dummy variable | Use the month as a time dummy variable to control the impact of release time changes on participation time. |

## 5. Empirical Results

The data, descriptive statistics, and correlation results are shown in Tables 2 and 3, respectively. From the descriptive statistics, we observed that the minimum participation time is less than one minute. This is because some solvers have done the task before registration and submission, so they could sign up and submit solutions in a short time. However, the proportion of this behavior is small in the overall sample, which could not affect the overall regression results, and the average participation time is 892.1 min, about 15 h, in line with the actual reality. From the correlation results, we can see that there is a certain correlation between some variables. For this, we tested the variance expansion factor between the variables and the results showed that they are all less than 5, which means there is no multicollinearity between the variables. Regression analysis is feasible among these variables.

**Table 2.** Descriptive statistic.

|  | Mean | Std | Min | Max | N |
|---|---|---|---|---|---|
| Participation Time | 892.1 | 1752 | 0.333 | 53,092 | 2500 |
| Task Rewards | 529.5 | 708.8 | 100 | 20,000 | 2500 |
| Task Duration | 29.21 | 45.66 | 0.19 | 728.2 | 2500 |
| Description length | 403.8 | 398.7 | 0 | 2419 | 2500 |
| Description Attachment | 0.234 | 0.423 | 0 | 1 | 2500 |
| Task Feedback | 1.205 | 1.677 | 0 | 19 | 2500 |
| Extra Explanation | 0.292 | 0.455 | 0 | 1 | 2500 |
| Task Density | 38.84 | 9.255 | 6 | 65 | 2500 |
| Market Price | 526.8 | 122.1 | 284 | 1357 | 2500 |
| Seeker's Credit | 15.85 | 24.47 | 0 | 210 | 2500 |
| Seeker's Experience | 3.892 | 6.232 | 1 | 67 | 2500 |
| Seeker's Authentication | 0.161 | 0.368 | 0 | 1 | 2500 |

**Table 3.** Correlation of variables.

|  | 1 | 2 | 3 | 4 | 5 | 6 | 7 | 8 | 9 | 10 | 11 | 12 |
|---|---|---|---|---|---|---|---|---|---|---|---|---|
| 1 Participation Time | 1 | | | | | | | | | | | |
| 2 Task Rewards | 0.2083 * | 1 | | | | | | | | | | |
| 3 Task Duration | 0.2044 * | 0.1057 * | 1 | | | | | | | | | |
| 4 Description length | 0.0708 * | 0.2460 * | 0.0275 | 1 | | | | | | | | |
| 5 Description Attachment | 0.0551 * | 0.0374 | −0.0027 | 0.0096 | 1 | | | | | | | |
| 6 Task Feedback | 0.2084 * | 0.3337 * | 0.1081 * | 0.0449 | 0.035 | 1 | | | | | | |
| 7 Extra Explanation | 0.1245 * | 0.046 | −0.0003 | 0.0139 | 0.2087 * | 0.1526 * | 1 | | | | | |
| 8 Task Density | −0.0132 | 0.0022 | −0.0085 | −0.0139 | 0.0046 | −0.0941 * | 0.0138 | 1 | | | | |
| 9 Market Price | 0.0492 | 0.1756 * | 0.0231 | 0.0581 * | −0.0185 | 0.0089 | 0.0012 | 0.0128 | 1 | | | |
| 10 Seeker's Credit | 0.1167 * | 0.4157 * | 0.1210 * | 0.0639 * | 0.0728 * | 0.0964 * | −0.0161 | 0.0049 | 0.0905 * | 1 | | |
| 11 Seeker's Experience | −0.0188 | −0.0446 | 0.1231 * | −0.0664 * | 0.0463 | −0.0624 * | −0.0401 | 0 | −0.012 | 0.6348 * | 1 | |
| 12 Seeker's Authentication | 0.0117 | 0.0238 | −0.0034 | −0.0221 | 0.0429 | 0.0496 | 0.0203 | 0.0064 | 0.0119 | 0.2321 * | 0.1710 * | 1 |

Note: The asterisk (*) indicates that the correlation coefficient is significant at a confidence level of 0.01.

We used STATA software to perform stepwise least squares regression on the samples. The final results are shown in Table 4. It should be noted that there are some null value samples in the task description variable, which were rejected during logarithmic processing. Column 1 of Table 4 reports the regression of task rewards and duration to participation time with a significant confidence level of the coefficient. Then, we added the quadratic term of task duration into the regression model and the result is reported in column 2. The sign of one term of task duration is positive and the quadratic term is negative, indicating that there is a significant inverted U-shaped relationship between the task duration and the participation time with the significance level $p < 0.001$. So, Hypothesis 1a and 1b are proved. Column 3 shows that we added the description length and description attachment and found that the relationship between the length of description and participation time is negative, with a significance level of $p < 0.001$. The relationship between description attachment and the participation time is positively significant, with significance level of $p < 0.001$. Thus, hypothesis 2a and 2b are proved. Column 4 shows that we added task communication and supplemented instructions with attachment and the results describe that both variables were positively related to participation time, thus, Hypothesis 3a and 3b are certified. Column 5 adds the task density and market price and it also concludes all the variables in this regression. The regression coefficient of the task density was negative, with a significance level of $p < 0.05$, and the regression coefficient of the market price was positive, with significance level of $p < 0.01$. Thus, hypothesis 4a and 4b are proved.

**Table 4.** Regression results.

| | (1) | (2) | (3) | (4) | (5) |
|---|---|---|---|---|---|
| | Ln(Participation Time) | Ln(Participation Time) | Ln(Participation Time) | Ln(Participation Time) | Ln(Participation Time) |
| Ln(Rewards) | 0.4673 *** | 0.4431 *** | 0.5063 *** | 0.4026 *** | 0.3845 *** |
| | (0.000) | (0.000) | (0.000) | (0.000) | (0.000) |
| Ln(Duration) | 0.7741 *** | 1.7746 *** | 1.8161 *** | 1.7806 *** | 1.7702 *** |
| | (0.000) | (0.000) | (0.000) | (0.000) | (0.000) |
| Ln(Duration)$^2$ | | −0.1717 *** | −0.1775 *** | −0.1771 *** | −0.1763 *** |
| | | (0.000) | (0.000) | (0.000) | (0.000) |
| Ln(Description length) | | | −0.1544 *** | −0.1333 *** | −0.1331 *** |
| | | | (0.000) | (0.000) | (0.000) |
| Description Attachment | | | 0.3812 *** | 0.2628 ** | 0.2684 ** |
| | | | (0.000) | (0.009) | (0.007) |
| Task Feedback | | | | 0.1208 *** | 0.1195 *** |
| | | | | (0.000) | (0.000) |
| Extra Explanation | | | | 0.4892 *** | 0.4889 *** |
| | | | | (0.000) | (0.000) |
| Ln(Task Density) | | | | | −0.4660 * |
| | | | | | (0.022) |
| Ln(Market Price) | | | | | 0.6566 ** |
| | | | | | (0.002) |
| Control variable | Yes | Yes | Yes | Yes | Yes |
| Time dummy variable | Yes | Yes | Yes | Yes | Yes |
| Constants | −0.0203 | −1.1148* | −0.7551 | −0.3831 | −2.6634 |
| | (0.960) | (0.011) | (0.096) | (0.399) | (0.082) |
| N | 2500 | 2500 | 2348 | 2348 | 2348 |
| R$^2$ | 0.186 | 0.199 | 0.207 | 0.225 | 0.229 |
| F | 35.3944 | 36.1969 | 32.0488 | 32.0923 | 30.0730 |

Note: For the *p*-values in parentheses, * means $p < 0.05$, ** means $p < 0.01$, *** means $p < 0.001$, and Yes indicates that control variables are included in the model.

## 6. Conclusions

### 6.1. Research Conclusions

This study analyzed the factors affecting the solvers' participation time in the crowdsourcing contest, which helps firms or organizations to obtain higher quality solutions for crowdsourcing contest. We choose four factors including task design, task description, task communication process, and market environment and studied their impacts on the participation time of solvers in crowdsourcing contests. Based on the literature review and related theories, the following conclusions were drawn by using the latest data from the Taskcn.com website.

First, Hypotheses 1a and 1b are verified by the empirical results in column 2 and we conclude that higher task rewards will increase the participation time of solvers, while task duration has an inverted U-shaped relationship with participation time, which suggests that the impact of the task duration on participation time is positive within a certain range. Until the duration exceeds a threshold, a longer task duration will increase the time cost of solvers and inversely reduce the participation time.

Second, Hypotheses 2a and 2b are verified by the results in column 3 and the length of the description released by seekers affects solvers' participation time. When the description length is too long, it has a negative effect on the participation time. Adding the description attachment often makes the description information clearer, and positively affects solvers to expend time to complete the task.

Third, Hypotheses 3a and 3b are verified by the column 4 and we conclude that communication and supplemental explanations during the task process can help reduce task uncertainty and reduce information asymmetry between seekers and solvers, thus it is positively affecting solvers' participation time.

Fourth, the Hypotheses 4a and 4b are verified by the column 5 and it shows that the greater the task density, the more choices the solver faces in the same time, the time allocated to each task will be reduced; when the market price is high, the solver will invest more time to participate in order to increase the expected value of winning.

### 6.2. Research Significance

This research mainly focuses on and contributes to the research field of online crowdsourcing contests, which takes the participation time referring to participation behavior as the dependent variable for the first time, further enriching the research on participation behavior and the factors of participation behavior. As the participation time can objectively reflect solvers' efforts during the participation process, focusing on the participation time is beneficial for our research on solver' behaviors. For this, we put forward some theoretical assumptions and models which are more realistic and conducted a scientific and objective empirical test. This has important significance for firms or organizations to understand how to increase the performance itself through task design and attracting more solvers to expend more participation effort.

### 6.3. Limitations and Future Research Directions

However, this study still has some limitations. First, the sample in this study is only subject to the logo design task. For different types of tasks, it was been discussed in depth. Whether the research results can be universally applied to any type of crowdsourcing task needs further discussion. Second, the study failed to consider the impact of the task difficulty itself on the behavior of solvers. As the difficulty level of the task itself has a strong subjectivity, it is difficult to measure through quantitative methods.

In future research, we will classify solvers with different characteristics and enrich the universality of research conclusions. We will also introduce the difficulty of the task and study the impact of task design on the participation time of solvers under the condition of controlling the difficulty of the task.

**Funding:** This research was funded by "National Key Research & Development Plan of China, grant number 2017YFB1400100" and "The National Natural Science Foundation of China was funded by 71702206".

**Conflicts of Interest:** The author declares no conflict of interest.

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
