# Peer review of "Research on Factors Affecting Solvers’ Participation Time in Online Crowdsourcing Contests"

_futureinternet, doi:10.3390/fi11080176_

Round 1

Reviewer 1 Report

Relevance to the themes of the Journal is very good.

The abstract is appropriate as a description of the paper

 The methodology is well described.

The quoted literature is abundant and helps the author's explanations to be clear and logical. Also the assumptions are well founded, supported by important articles.
The study is complex and the conclusions are consistent.

Reference list is adequate and correctly cited.

Standard of English is good.

The suggestion would be to provide more information (link) about the site Taskcn.com involved in the article as the source of the processed data because it can not be accessed.

Author Response

Dear reviewer:

Thank you very much for your comments and suggestions to the article. I have revised my manuscript point by point according to your comments and use the word revision mode when modifying. You can check it out in the revised version.

1.English language and style are fine/minor spell check required

Answer: I have corrected and modified the English language and style of the paper.

2.The Taskcn.com cannot be accessed.

Answer: www.Taskcn.com is the only address for taskcn.com. I have visited this website last month and access to it. But recently I also tried to visit it again and I found that I also could not access it. So, I contacted the customer service of the company and they said that the website is being updated. The access time has not been determined. I am very sorry for this. Thus, in order to prove the authenticity and validity of my paper, it is proved from the website screenshots, data screenshots and the existing research reference based on Taskcn.com data.

Thank you very much for your comments and suggestions.

Best regards

Reviewer 2 Report

Thank you for the opportunity of reviewing your article!

It is an interesting article, well written and addressing a topic of high interest for scholars and practitioners. The manuscript is inspiring and innovative, but, at the same time meticulous and accurate in its approach.

I would have some considerations and suggestions for improving the quality of the article

The author states a series of hypotheses (both principal and derived), and aims to analyse them within the research proper. The paper would benefit if, in the final section (Discussions), the author would explicitly mention the results for each hypothesis (acceptance, rejection, inconsistent results), and then follow up with general conclusions.

Brief final considerations on integrating the paper within the framework of existing literature on the topic (confirmations and disputed results, with respect to other papers) would be beneficial.

Formal comments

-        Review formatting (i.e ... review above, We believe that... row 189, 

-        Diferent fonts and sizes

-        Review in-line references (see Li and Hu (2017) or Ye and Kankanhalli (2017) page 3, instead of Li and Hu [17] , or Ye and Kankanhalli [18] and others, mostly in the Literature review section

-        Same for References section – journal abbreviation, according to mdpi style and instructions

Thank you for the opportunity to review this article and good luck!

Author Response

Dear reviewer

Thank you very much for your valuable comments. I have revised the issues in the text one by one according to your comments. The main changes are as follows:

Point 1: English language and style are fine/minor spell check required 

Response 1. Corrected and modified the English writing of the paper. For the modification, please refer to the revised version submitted.

Point 2: Review formatting (Diferent fonts and sizes...)

Response 2: Corrected the font and format of the manuscript.

Point 3: Reference format  (see Li and Hu (2017) or Ye and Kankanhalli (2017) page 3...)

Response 3: For your comments on the reference format, I have done some revision. The journal abbreviations in the reference section have also been revised.

Point 4: The paper would benefit if, in the final section (Discussions), the author would explicitly mention the results for each hypothesis (acceptance, rejection, inconsistent results), and then follow up with general conclusions.Brief final considerations on integrating the paper within the framework of existing literature on the topic (confirmations and disputed results, with respect to other papers) would be beneficial.

Response 4: For your suggestions for the conclusion section, I have revised it and use the word revision mode when modifying. You could check it out in the revised manuscript.

Thank you very much for your comments and suggestions.

Best regards
